# TRPV1: A Common Denominator Mediating Antinociceptive and Antiemetic Effects of Cannabinoids

**DOI:** 10.3390/ijms231710016

**Published:** 2022-09-02

**Authors:** Kathleen Louis-Gray, Srinivasan Tupal, Louis S. Premkumar

**Affiliations:** 1Department of Neurology, University of Michigan, Ann Arbor, MI 48109, USA; 2Department of Pharmacology, Southern Illinois University School of Medicine, Springfield, IL 62794, USA

**Keywords:** cannabinoids, Δ^9^-tetrahydrocannabinol, cannabidiol, TRPV1, resiniferatoxin, antinociception and antiemesis

## Abstract

The most common medicinal claims for cannabis are relief from chronic pain, stimulation of appetite, and as an antiemetic. However, the mechanisms by which cannabis reduces pain and prevents nausea and vomiting are not fully understood. Among more than 450 constituents in cannabis, the most abundant cannabinoids are Δ^9^-tetrahydrocannabinol (THC) and cannabidiol (CBD). Cannabinoids either directly or indirectly modulate ion channel function. Transient receptor potential vanilloid 1 (TRPV1) is an ion channel responsible for mediating several modalities of pain, and it is expressed in both the peripheral and the central pain pathways. Activation of TRPV1 in sensory neurons mediates nociception in the ascending pain pathway, while activation of TRPV1 in the central descending pain pathway, which involves the rostral ventral medulla (RVM) and the periaqueductal gray (PAG), mediates antinociception. TRPV1 channels are thought to be implicated in neuropathic/spontaneous pain perception in the setting of impaired descending antinociceptive control. Activation of TRPV1 also can cause the release of calcitonin gene-related peptide (CGRP) and other neuropeptides/neurotransmitters from the peripheral and central nerve terminals, including the vagal nerve terminal innervating the gut that forms central synapses at the nucleus tractus solitarius (NTS). One of the adverse effects of chronic cannabis use is the paradoxical cannabis-induced hyperemesis syndrome (HES), which is becoming more common, perhaps due to the wider availability of cannabis-containing products and the chronic use of products containing higher levels of cannabinoids. Although, the mechanism of HES is unknown, the effective treatment options include hot-water hydrotherapy and the topical application of capsaicin, both activate TRPV1 channels and may involve the vagal-NTS and area postrema (AP) nausea and vomiting pathway. In this review, we will delineate the activation of TRPV1 by cannabinoids and their role in the antinociceptive/nociceptive and antiemetic/emetic effects involving the peripheral, spinal, and supraspinal structures.

## 1. Introduction

The effects of cannabis have been known since 2737 BC, when the Chinese emperor, Shen-Nung, used it to treat symptoms associated with gout. Since then, there have been many claims of the beneficial effects of cannabis, while at the same time, studies have raised skepticism regarding its usefulness as an antinociceptive and antiemetic agent [1,2]. In this review, we will focus on the transient receptor potential vanilloid 1 (TRPV1) ion channel and its interactions with cannabinoids.

Marijuana (cannabis) refers to both the whole marijuana plant as well as a raw, unprocessed preparation. Often, the term “medical marijuana” is used, but these preparations are not actually approved by the FDA as medicines. The identified phytochemicals are called cannabinoids. Although there are more than 450 phytochemicals in this plant, the major components are Δ^9^-tetrahydrocannabinol (THC) and cannabidiol (CBD) in the Cannabis sativa species, and the hemp plant also has high concentrations of CBD [3,4].

Cannabis has been used for centuries for recreational purposes because of its psychoactive properties. It has been well-characterized that THC exerts its effects on specific receptors, which have been identified as cannabinoid receptor 1 (CBR1) and cannabinoid receptor 2 (CBR2). CBR1 is distributed in the peripheral and central nervous systems, and the central CBR1 mediates the psychoactive effects of THC. The effects of CBR1 are mediated by G-protein-coupled receptor pathways [5,6,7]. Meanwhile, CBR2 is primarily expressed in the immune system and the nervous system and it modulates responses via G-protein-coupled receptors as well [5,6,7].

CBD has no psychoactive properties and has been utilized in the treatment of certain childhood epilepsies [8,9]. Unlike THC, a specific receptor for CBD has not been identified, but there are studies showing that it interacts with TRPV1 to exert some of its effects [10,11,12]. The FDA has also approved a CBD-based liquid medication, Epidiolex (cannabidiol), which has been used for the treatment of two forms of severe childhood epilepsy, Dravet syndrome and Lennox-Gastaut syndrome [13]. This effect is specific to formulations containing higher amounts of CBD. It appears that CBD reduces both the frequency and the severity of the episodes. There is evidence that TRPV1 may be involved in febrile epilepsy [14].

TRPV1, a nonselective cation channel with a high Ca^2+^ permeability is expressed in the small-diameter sensory neurons and supraspinally in the central descending pain pathways that regulate nociception [15,16,17,18,19,20,21,22]. In fact, activation of TRPV1 in the ascending pain pathway mediates nociception, whereas activation of TRPV1 in the descending pain pathway mediates antinociception.

Several transient receptor potential (TRP) channels have been cloned and characterized. There are 6 families of TRP channels: TRP Vanilloid 1–4 (TRPV1–4), TRP Canonical (TRPC), TRP Melastatin (TRPM8), TRP Ankyrin (TRPA), TRP Polycystin (TRPP), and TRP Mucolipin (TRPML) [23,24,25]. TRPV1 channels are activated by phytochemicals, such as capsaicin, an ingredient in hot chili peppers (*Capsicum annuum* or *frutescens*), and by resiniferatoxin (RTX), which can activate TRPV1, which is obtained from a spurge (*Euphorbia resinifera*/*poissonii*) [15,16,26,27,28]. Interestingly, cannabinoids obtained from *Cannabis sativa/indica/ruderalis* have been shown to activate TRPV1; however, the major psychoactive cannabinoid Δ^9^-tetrahydrocannabinol (THC) does not activate TRPV1, whereas the other major cannabinoid, cannabidiol (CBD) is a potent activator of TRPV1 [9,10,11,12]. Other minor cannabinoids have also been shown to activate TRPV1 [10]. Cannabigerol (CBG) is reported to act as a ligand for TRPV1 [10]. THC has been shown to potently activate TRPV2 [10,29]. A widely used CBR1 agonist, WIN55, 212-2, has been shown to exert some of its effects through the activation of TRPV1 and TRPA1 [30,31,32]. Also, it should be considered that ajulemic acid is a metabolite of THC which shows anti-inflammatory and antifibrotic effects without exerting psychoactive properties [33]. Tetrahydrocannabivarin (THCV and THV) is a homologue of THC which acts as an antagonist of CBR1 [5]. Anandamide (AEA) is an endocannabinoid that activates both CBR1 and TRPV1 receptors [34,35,36].

There are complex effects resulting from interactions between the effects of THC and CBD. The actions of CBD on TRPV1 also have an impact on this interaction. Chronic inflammatory pain is mediated by the sensitization of TRPV1 by various mechanisms, including its phosphorylation [37,38,39]. Since CBR1 receptors are negatively coupled to cAMP via Gi, CBR1-mediated dephosphorylation of TRPV1 may indirectly affect the downstream effects of TRPV1 by CBD.

One example of the interplay between THC and CBD includes the way in which CBD reduces the psychosis-like effects of THC. There are studies suggesting that CBD may have its own antipsychotic effects [40,41,42,43,44]. In animal models, acute exposure to THC affects cognitive behavior; it induces dose-related effects on decision making, abstract-thinking abilities, and executive functions [45,46,47]. The most remarkable effects are on the short-term working memory, verbal skills, and attention deficits [45,46,47,48,49,50,51]. Interestingly, CBD is able to reduce THC-induced cognitive impairment [43]. As discussed earlier, most preparations contain unknown amounts of THC and CBD; therefore, the discrepancies found in studies as to the effects of CBD could be attributed to these interactions.

Cannabis is also used as an antiemetic agent in various conditions. One of the side effects of the chronic and improper use of cannabis is hyperemesis syndrome (HES), which is becoming more common with increased use as a result of the legalization of marijuana [52,53,54,55]. Nausea and vomiting result from complex interactions between afferent and efferent pathways of the gastrointestinal tract, central nervous system, and autonomic nervous system [52,53]. The role of TRPV1 has been demonstrated by experiments conducted with resiniferatoxin (RTX), a potent TRPV1 agonist. Lower concentrations of RTX act as an antiemetic, while higher concentrations induce emesis. The use of CBD alone, via the activation of TRPV1, is likely to induce antiemetic effects [10]. However, when TRPV1 is downregulated by the activation of CBR1-receptor-mediated dephosphorylation, this could result in HES [6,7]. As indicated above, the combination ratios of THC and CBD are critical. Specific formulations have been approved by the FDA, such as dronabinol (Marinol) and nabilone (Cesamet). These contain THC as the active ingredient, which can be useful in treating chemotherapy-induced nausea and as an appetite stimulant [56].

Given the outcomes of recent legislation, it is likely that marijuana will be legalized in many states in the U.S. and in other parts of the world in the future [54,55]. Therefore, rigorous scientific research must be conducted so as to identify the specific targets, pharmacological effects, and the pharmacokinetic and pharmacovigilance profiles for the effective use of cannabis, and that should be supported by evidence-based clinical trials. The purported uses of cannabinoids are for relief of pain and for prevention of nausea and vomiting. The mechanisms underlying these effects are not fully understood. In this review, we will delineate the role of TRPV1 in inducing these antinociceptive and antiemetic properties. We will also discuss the role of TRPV1 in the possible reduction of the antinociceptive effect and in avoiding hyperemesis syndrome (HES) following the use of cannabinoids.

## 2. Role of TRPV1 in Cannabinoid-Induced Antinociception

Pain is carried from the periphery by nociceptors Aδ and C-fibers, which are thinly myelinated and unmyelinated, respectively. These fibers are further subdivided by their sensitivities to physical stimuli: the C-fibers that are responsible for sensing mechanical and heat stimuli are classified as CMH fibers, and there is also a set of C-fibers which is mechano-insensitive, classified as CMi fibers [56,57]. These nociceptors are TRPV1-expressing peptidergic (CGRP and substance P (SP)-releasing) fibers. Neuropathic pain occurs as a result of the abnormal activity of Aδ and C nociceptors, which is associated with several conditions, such as peripheral nerve injuries [58], painful DPN [59,60], painful peripheral herpes neuropathy (PHN) [61,62], painful HIV-associated neuropathy (HIV-AN) [63], complex regional pain syndrome (CRPS) [64], small-fiber neuropathy in metabolic syndrome [65], neuropathic pain manifestations in Fabry disease [66], and chemotherapy-induced peripheral neuropathy [67,68]. Cannabis use has been claimed to be useful in relieving pain in these conditions [1,2,69]. In a small number of patients with HIV-AN, one study showed a 30% reduction in reported pain after a week of smoking medicinal cannabis [70].

The classic behavioral effects of cannabis in rodents are called a “tetrad”, which includes the reduction in body temperature, analgesia, reduced locomotion, and catalepsy [71]. Hypothermia could be explained by the activation of TRPV1, and analgesia could be explained by the desensitization/downregulation of peripheral TRPV1 or by the activation of central TRPV1 in the descending pain pathway [72,73]. The interactions of CBD and THC could occur through TRPV1 channels. CBD potentiated the suppression of locomotion and reduced hypothermia caused by THC when administered in a 1:1 (CBD:THC) ratio in mice [74], but it potentiated both the suppression of locomotion and hypothermia when administered in a 10:1 (CBD:THC) or a 50:1 (CBD:THC) ratio [75]. Administration of CBD in rats (20 mg/kg, i.p.) prolonged the duration of hyperthermia and hypolocomotion [76]. These studies suggest interactions between THC and CBD, as well as between TRPV1 and CBD.

TRPV1 channels are expressed in the peripheral nerve terminals of nociceptors (Aδ and C-fibers). Upon activation, it depolarizes the nerve terminals, generates action potentials, and propagates noxious information to the higher brain centers via the spinal cord [15,21,77,78,79]. Activation of TRPV1 can also cause the release of neuropeptides, such as CGRP and SP, from the peripheral and central nerve endings. CGRP is known to be a potent vasodilator [80,81]. Blood vessels are strongly stained for TRPV1 [82]. Cannabinoids have been shown to exert powerful hypotensive effects [83]. TRPV1 channels are also expressed in the central nerve terminals of sensory neurons, where peripheral afferents form synapses at the spinal dorsal horn, vagal nerve afferents at the NTS, and trigeminal nerve afferents at the caudal trigeminal nucleus [84,85,86,87,88,89,90]. TRPV1 channels are expressed in specific locations in the higher centers of the brain, such as the descending pain pathway involving the rostral ventral medulla (RVM) and the periaqueductal gray (PAG) [91,92,93,94,95,96]. However, there are controversies regarding the extent of TRPV1 expression in the central nervous system. Studies have shown that its expression is restricted to the peripheral nervous system [82]. TRPV1 channels expressed in the periphery mediate nociception, whereas TRPV1 channels expressed in the descending pain pathway mediate antinociception. The sustained activation of these channels at nerve terminals can cause the desensitization/depolarization block of the nerve terminals, preventing the generation and propagation of action potentials. RTX, an ultrapotent agonist of TRPV1, is very effective in inducing the depolarization block resulting in antinociception [26,27,28].

In order to account for the interaction, the effects of phosphorylation on TRPV1 must be taken into consideration. PKC- and PKA-mediated phosphorylation robustly potentiates TRPV1 functions by sensitizing the receptors, and thereby the nociceptors (C and Aδ fibers) [16,37,38,39]. This effect underlies the basis for inflammatory pain and the development of TRPV1 antagonists as analgesic and anti-inflammatory agents to treat certain modalities of pain. However, in clinical trials, it became apparent that the TRPV1 blockade increased the core body temperature, which led to the abandonment of developing TRPV1 antagonists as analgesics [97]. Also, it should be taken into consideration that the blockade of central TRPV1 in the descending pain pathway mediates antinociception [91,92,93,94,95,96]. In regard to cannabinoids, if CBR1 and TRPV1 are expressed in the same neuron, the CBR1-mediated reduction in cAMP levels could downregulate TRPV1 expression and function by reducing the phosphorylated state of the channel. Therefore, when a mixture of THC and CBD in a preparation is consumed or administered, depending upon the quantities of each of the active ingredients, the activation of TRPV1 by CBD and the downregulation of TRPV1 by the activation of CBR1 could mutually nullify the effects [98,99,100,101,102]. Some of the well-known formulations, such as nabiximols, have a combination of THC and CBD [103,104]. Another added complexity is that some of the minor cannabinoids, such as THCV, act as antagonists of CBR1 [10]. A careful analysis should be carried out using pure ingredients with known proportions in the mixtures to delineate the ultimate effects of cannabinoids [104]. The United Kingdom, Canada, and several European countries have approved nabiximols (Sativex), which is an equal ratio (1:1) mixture of THC (2.7 mg) and CBD (2.5 mg), formulated as a mouth spray to alleviate neuropathic pain, incontinence, spasticity, and multiple sclerosis, but it has not been approved by the FDA. It should be mentioned that CBD is freely available for online purchase from several companies in different concentrations accompanied by analytical data regarding possible contaminants, including pesticides [105].

Despite decades of research and clinical usage of cannabis in treating chronic pain conditions, incontrovertible evidence for its efficacy has yet to be established [106]. This may be due to the lack of a clear understanding of the mechanism of action and the use of unregulated combinations of cannabinoids in treatment regimens [10,103,104]. Changes in legislation are being promulgated in different countries; therefore, it is likely that use will increase worldwide. These products are most commonly used for chronic pain conditions—apart from their recreational use [107].

Chronic pain is considered to be the most significant cause of disability globally. Several preclinical and clinical studies have been undertaken, but the results are conflicting. Some studies show significant effects, and other show minimal effects. A review has been compiled recently using the outcomes specified in the Initiative on Methods, Measurement, and Pain Assessment in Clinical Trials (IMMPACT), which quite controversially concludes that it is unlikely that cannabinoids are effective as medicine for treating chronic noncancer pain [106].

Capsaicin, a competitive TRPV1 agonist, when applied peripherally, induces intense burning pain. It has been shown that TRPV1 expression and function are increased and decreased in hyper- or hypoalgesia, respectively, in the periphery and at the first sensory synapse in the spinal dorsal horn [108]. The role of TRPV1 has been confirmed using a potent TRPV1 agonist, RTX, that induces a depolarization block by persistently activating TRPV1 expressed in the nerve terminals in the short-term, as well as nerve terminal desensitization/depletion in the long-term and reversed thermal hyperalgesia [26,27,108]. RTX is currently in clinical trials for the treatment of certain terminal-cancer pain conditions [109,110,111,112,113,114,115,116].

Pain involving the head and neck is carried by trigeminal neurons. Trigeminal neurons that synapse at the caudal trigeminal nucleus express TRPV1, and TRPV1-mediated CGRP, a potent vasodilator, release has been implicated in migraine-type headaches [117,118], and cannabis has been shown to be effective in the treatment of migraines. In a study, 11% of migraineurs reported complete resolution [119]. Given the knowledge we have gained regarding TRPV1 and pain, and also that regarding TRPV1 and CGRP levels in migraine, it is conceivable that cannabinoids could play a role in migraines. The activation of CBD might worsen the symptoms as a result of a further increase in TRPV1-mediated CGRP-release, causing the vasodilation of the meningeal vessels, while on the other hand, THC activation of CBR1 may downregulate TRPV1 function and decrease CGRP-release and aid in the relief of migraine pain. It has been suggested that cannabinoids may be useful in relieving craniofacial pain associated with dental problems, anxiety, trigeminal neuralgia, and temporomandibular joint dysfunction [120,121,122].

Our bodies also produce their own cannabinoid chemicals, called “endocannabinoids”. They play a role in regulating pleasure, memory, thinking, concentration, body movement, awareness of time, appetite, pain, and the senses (taste, touch, smell, hearing, and sight). Neuronal activity increases the production of endovanilloids/cannabinoids during chronic pain conditions and play a role in anxiety [123,124,125]. Endovanilloids, such as N-arachidonyl ethanolamine (anandamide, AEA) and 2-arachidonyl glycerol (2-AG), are synthesized and released on demand and metabolized by fatty acid amide hydroxylase (FAAH) and monoacylglyceride lipase (MAGL), respectively [126,127,128]. The activation of CBR1 and CBR2 receptors stimulates the production of oleoylethanolamide (OEA), which is known to activate peroxisome proliferator-activated receptor-α (PPAR-α) [129], which is involved in fat metabolism; therefore, it could be useful in treating obesity [130]. Interestingly, the activation of TRPV1 by OEA enhances metabolism in brown fat cells [131].

Liao et al., 2011 [132] have concluded from their experiments in the PAG that the activation of TRPV1 in the glutamatergic terminals releases glutamate, which in turn activates the metabotropic glutamate receptors (mGluR) in the postsynaptic cells. The activation of mGluRs is involved in the synthesis of 2-AG, which retrogradely activates CBR1 in the presynaptic terminal in ventrolateral PAG (vlPAG), causing reduced gamma aminobutyric acid (GABA) release and mediating antinociception in the descending pain pathway [133,134,135].

PAG is a midbrain structure whose role in the descending control of pain has been quite well-established. Glutamatergic neurons project from vlPAG to the adjacent RVM. Physiologically, this system tends to suppress pain signals [133,134]. Three kinds of neurons (ON-, OFF- and neutral cells) from the RVM project to the dorsal horn of the spinal cord. The OFF-cells in the RVM can be stimulated by the glutamatergic axonal projections from the vlPAG [135,136,137]. Any intervention that increases OFF-cell firing in the PAG or the RVM inhibits the transmission of pain signals from the periphery. On the contrary, ON-cell activity facilitates pain transmission [137,138,139]. One of the most studied central analgesic mechanisms of cannabinoids and opiates is the disinhibition of vlPAG glutamatergic-projecting neurons that results in the stimulation of RVM OFF-cells. The mechanism of disinhibition by these agents is the attenuation of GABA release from interneurons in the PAG [140,141].

Increased glutamatergic transmission in the projecting neurons in vlPAG induces antinociception. The administration of the TRPV1 agonist capsaicin into vlPAG increases glutamatergic neuronal firing and increases GABA release. However, CBR1-receptor activation by WIN55, 212-2 induces antinociception by decreasing GABA-release, leading to the disinhibition of the glutamatergic neurons. In painful diabetic peripheral neuropathy, the TRPV1-mediated antinociceptive effect is attenuated as a result of the reduction of the expression of the TRPV1 receptors. On the other hand, when CBR1-receptor expression is increased, that could lead to a reduction in GABA release, thereby further augmenting the disinhibition of the glutamatergic transmission [92].

Administration of intra-vlPAG palmitoylethanolamide (PEA), a PPAR-α agonist, induces antinociceptive effects, which are seen as a decrease in the RVM ON- and OFF-cell activities. PPAR-α responses are mediated by the activation of the CBR1 and TRPV1 receptors. The TRPV1 blocker, iodo-RTX, had no effect on the ON-cell activity or tail-flick latency, whereas it blocked the PEA-induced decrease in the ongoing activity of the OFF-cell [142]. The roles of CBR1 and TRPV1 receptors have been further confirmed by blocking the degradation of AEA (which activates both the CBR1 and TRPV1 receptors) by FAAH inhibitor, which revealed the modulation of synaptic transmission in the PAG [143,144]. It has been shown that, in cells expressing CBR1 and TRPV1 receptors, the activation of the CBR1 receptor either stimulates or inhibits, and the inhibition is dependent upon the activation of cAMP signaling [145]. dlPAG determines the core affective aspects of aversive memory formation controlled by the local TRPV1/CBR1 balance [146].

5-HT1A, an autoreceptor of serotonin expressed presynaptically, can modulate neurotransmitter release. If the coupling mechanism is similar to that of CBR1, TRPV1 receptors expressed in the vicinity could be downregulated. CBD administration into dlPAG has been shown to decrease anxiety, which is mediated by 5-HT1A receptors, but not by CBR1 because a CBR1-receptor antagonist (SR141716 or SR144528) had no effect [147]. The dorsal raphe nucleus (DRN) is involved in nociception [20,148,149,150,151]. There are reports that, following the induction of neuropathic pain, 5-HT neurons in DRN show a decrease in firing rate [152], but other studies have shown an increase in neuronal firing [153]. In animal models of neuropathic pain, CBD is able to reverse mechanical allodynia, but not the antianxiety effects mediated by the 5-HT1A receptors which could be reversed by TRPV1 antagonists [10,11,154].

The pain relief could be associated with psychoactive properties of cannabinoids [155,156]. Combinations of THC + CBD have been useful to reduce anxiety and could be useful in generalized social anxiety disorder. It has been reported that CBD alone could be useful in reducing anxiety and cognitive impairment [156]. Some of these effects could be related to the concentration of THC and CBD in a given preparation. The effects of CBD have been compared to a known antidepressant drug, imipramine [157]. This observation raises the question as to whether cannabinoids cause depression. In an elaborate study involving 6900 subjects, with an age ranging from adolescent to mature adult, there was no indication that cannabinoids cause depression [158,159,160].

Multiple sclerosis (MS) is a debilitating condition; it causes neuronal inflammation and muscle spasticity. MS patients benefit from use of cannabis [161]. Two patient surveys have revealed that, in spinal cord injuries, 50% of respondents reported that marijuana reduced muscle spasticity, and 97% of MS patients who used cannabis in conjunction with their therapy reported that cannabis improved spasticity, chronic pain, tremor, weight loss, and other symptoms [162].

## 3. Role of TRPV1 in Cannabinoid-Induced Antiemesis

One of the uses of cannabis is to prevent nausea and vomiting in various conditions. It is useful in improving appetite, probably acting as an antiemetic agent. There is widespread use of cannabinoids as antiemetic agents, especially during cancer chemotherapy [52,53]. Antiretroviral therapy for HIV/AIDS treatment results in a number of side effects, such as neuropathic pain, lack of appetite, anxiety, and depression. In this group of patients, cannabis has been shown to improve quality of life [163]. Although it has been shown that cannabis could be useful as an antiemetic, it can be also proemetic [52]. It is not recommended for use against pregnancy-induced nausea because the effects of cannabis on the unborn have not yet been established [164]. However, chronic use of higher levels of cannabinoids induces hyperemesis syndrome (HES). Antiemesis and HES caused by cannabinoids could be explained by their actions on TRPV1 channels in the emesis pathway [52,53].

The precise neurocircuitry involved in nausea and vomiting is not fully understood. It involves structures within the medullary reticular formation of the hindbrain, which includes the AP, NTS, and dorsal motor nucleus of the vagus (DMV) [160,165,166]. Although the effects are attributed to THC, there is increasing evidence that CBD may play a role in nausea and vomiting [167]. There is a clear link between TRPV1 and nausea and vomiting. Since CBD activates TRPV1, there could be an interaction. Given these findings, it is possible that chronic nausea caused by cannabinoids could be mediated via central TRPV1, similar to the antinociceptive effects of central TRPV1 in the descending pain pathway.

Therefore, it is necessary to understand the correlation between the activation of TRPV1 by CBD and its role in nausea and vomiting. Nausea and vomiting involve complex interactions between the afferent and efferent pathways of the gastrointestinal tract, the central nervous system, and the autonomic nervous system. Afferents from the vagus nerve, vestibular system, and chemoreceptor trigger- zone project to NTS, which in turn relays signals to initiate multiple downstream pathways mediating nausea and vomiting [165,168]. There appears to be a distinction between acute and chronic nausea; acute nausea originates from the GI tract in response to the consumption of toxic substances, whereas chronic nausea originates from the central neuronal circuits that can be equated to centrally mediated chronic neuropathic pain [169]. Several neuromodulators, including cannabinoids, have been shown to be efficacious in the treatment of nausea and vomiting. It is noteworthy that conventional antiemetic therapies used for the treatment of acute vomiting are not effective in treating chronic vomiting, suggesting disparate mechanisms [170].

However, long-duration, excessive use of cannabis can lead to HES, which is characterized by symptoms of cyclic abdominal pain, nausea, and vomiting. Hot-water hydrotherapy is a mainstay self-treatment for cannabinoid-induced HES, suggesting that the heat-induced activation of TRPV1 may play a role in the antiemetic effect [53,171,172,173]. Furthermore, topical capsaicin is a treatment option for HES, further suggesting the role of TRPV1. The downstream signaling pathway include the vagus nerve, NTS, and AP, acting via the SP/NK1 receptors [174].

Gut–brain signaling via the vagal nerve mediates motor functions and emesis, which could involve cannabinoid and TRPV1 receptors [175]. A similar mechanism that mediates nociception/antinociception via the activation of TRPV1 could also be involved in mediating the emetic/antiemetic effects of cannabinoids. The activation of TRPV1 by cannabinoids in the central emesis pathway may mediate antiemetic effects [176,177,178,179]. Similar to the effects in the central descending pain pathway, chronic use of higher concentrations of THC-containing mixtures of cannabinoids may downregulate TRPV1 via the activation of CBR1, resulting in an emetic response.

The role of TRPV1 has been confirmed by experiments conducted with RTX, a potent TRPV1 agonist. RTX is one of the most potent emetic substances described so far in animal models, mediated by TRPV1 located on neurons in the brainstem containing substance P [53,179,180]. However, RTX has also been shown to induce antiemetic effects. Very low concentrations of RTX irreversibly activate TRPV1, leading to depolarization block, but at higher concentrations, it leads to the desensitization/depletion of TRPV1-expressing nerve terminals, which may explain the concentration-dependent opposing effects of RTX [26,27,53]. Therefore, a dual effect could be expected with RTX; at lower concentrations, it is antiemetic, but at higher concentrations, it induces emesis. The downstream mechanism may involve the release of CGRP and SP and the activation of their respective receptors [80,81]. As discussed earlier, RTX is a unique compound, in that it can induce depolarization block by irreversibly activating TRPV1, resulting in the gradual inactivation of sodium channels, resulting in a failure to generate action potentials. Several TRPV1 agonists, such as arvanil, arachidonamide, AEA (which activates both CBR1 and TRPV1), and N-arachidonoyl-dopamine (NADA), all induce antiemetic effects [175]. Since CBD is known to activate TRPV1 [10], it is expected that it has antiemetic properties. However, constant, high-dose cannabis use results in HES. This is likely due to TRPV1 downregulation/desensitization via constant activation by CBD, as well as by CBR1 mediated by TRPV1 hypofunction. The area responsible for the action appears to be NTS, where vagus nerve terminals express TRPV1 and modulate synaptic transmission [179,180,181,182].

The involvement of substance P and its receptor, tachykinin (NK1), has been demonstrated by the direct application of SP to the dorsal brainstem in the AP, which induced emesis in ferrets [179]. This suggests that TRPV1-mediated CGRP and SP release from nerve terminals play a role in emesis [53]. NK1-receptor antagonists have been successfully used along with 5-HT3-receptor antagonists to treat chemotherapy-induced emesis [183,184]. Morphine and 8-hydroxy-2-(di-n-propylamino)tetralin (8-OH-DPAT) act as antiemetics [185]. The role of TRPV1 in the descending pain pathway has been established; in fact, antinociception induced by TRPV1 activation in the descending pathway involves the release of encephalin and endorphin at the level of RVM and PAG [91,92,93,94,95,96]. A similar mechanism could exist at the brainstem area to cause endorphin/encephalin-release-mediating antiemetic effects.

## 4. Concluding Remarks and Future Directions

The activation of the central TRPV1 channels by cannabinoids (CBD, but not THC) in the descending pain pathway that involves RVM and PAG plays a key role in the antinociceptive effects of cannabinoids. However, when CBR1 is also activated, the TRPV1 receptor is downregulated via a Gi-coupled mechanism by inhibiting protein kinase A-mediated phosphorylation and impairs the antinociceptive effects. Similarly, the activation of TRPV1 channels by cannabinoids in the NTS and AP circuitry mediates the antiemetic effects, whereas the downregulation of TRPV1 in this circuitry could result in HES. Therefore, maintenance of the central TRPV1 function is critical for mediating the analgesic effect and mitigating the adverse effect of HES.

## Data Availability

Not applicable.

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
