# Peer review of "TRPV1: A Common Denominator Mediating Antinociceptive and Antiemetic Effects of Cannabinoids"

_ijms, 2022, doi:10.3390/ijms231710016_

Round 1

Reviewer 1 Report

The review article by Louis-Gray et al. summarizes the actions of cannabinoids on the TRP channel and discuss a clear-cut link of cannabinoids induced activation/dephosphorylation of TRPV1 and its role in antinociceptive/nociceptive and antiemetic/emetic effects. Although this review is of good quality and well-written, it can be further improved after considering the below points:

1.     The authors may include a figure containing diagram of TRPV1 expression and its function. They may also include a schematic diagram for their conclusion, depicting the effects of activation of TRPV1 by cannabinoids and the effects of downregulation of TRPV1 due to CBR1 activation. This will be helpful to the readers, who are not familiar with this field, to easily understand the message of this article.

2.     Introduction section should be concise and re-organized. The authors may include a separate section on TRPV1 channel or the discussion on Page 3,4, Lines 134-151 should be moved to the Introduction section.

3.     Page 3, Line 135: ‘There are six families of TRP channels:’===> only 5 are mentioned, TRP mucolipin (TRPML) is missing.

4.     There are many places where font size is bigger. For examples: Page 2, line 87; Page 3, Line 138-141, Page 7, Line 334, 339

5.     Many references are not correctly formatted. The authors should check them thoroughly.

6.     Page 2, line 59 ‘which has bene used’ =>> ‘which has been used’

7.     Page 3, line 117 ‘neuropathy (PHN) 116 [49.50]’=>>>neuropathy (PHN) 116 [49,50]

8.     Page 3, Line 145 ‘CBG is reported to act as a ligand’=>> Is it CBD?

Author Response

Reviewer 1

We thank the reviewer for their useful comments on this manuscript. We have addressed all the issues to the best of our abilities point by point. I hope the reviewers find the answers adequate and the manuscript suitable for publication. We have included two files (changes shown and changes accepted)

  1. The authors may include a figure containing diagram of TRPV1 expression and its function. They may also include a schematic diagram for their conclusion, depicting the effects of activation of TRPV1 by cannabinoids and the effects of downregulation of TRPV1 due to CBR1 activation. This will be helpful to the readers, who are not familiar with this field, to easily understand the message of this article.

We have added a schematic diagram as suggested by the reviewer. The diagram does not show up in this box.

  1. Introduction section should be concise and re-organized. The authors may include a separate section on TRPV1 channel or the discussion on Page 3,4, Lines 134-151 should be moved to the Introduction section.

We have moved this section as suggested by the reviewer

  1. Page 3, Line 135: ‘There are six families of TRP channels:’===> only 5 are mentioned, TRP mucolipin (TRPML) is missing.

We have added TRPML TRP mucolipin to the list of TRP channels.

  1. There are many places where font size is bigger. For examples: Page 2, line 87; Page 3, Line 138-141, Page 7, Line 334, 339

We have corrected the font size.

  1. Many references are not correctly formatted. The authors should check them thoroughly.

We have checked the references for formatting errors.

  1. Page 2, line 59 ‘which has bene used’ =>> ‘which has been used’

We have corrected ‘bene’ to ‘been’

  1. Page 3, line 117 ‘neuropathy (PHN) 116 [49.50]’=>>>neuropathy (PHN) 116 [49,50]

Not clear what the change the reviewer is asking for.

  1. Page 3, Line 145 ‘CBG is reported to act as a ligand’=>> Is it CBD?

Cannabigerol (CBG) is a minor cannabinoid.

Reviewer 2 Report

The Authors presented a review describing the role of Transient Receptor Potential Vanilloid 1 (TRPV1) ion channel in the antinociceptive, antiemetic effects of cannabinoids. The opposing antinociceptive/nociceptive and antiemetic/emetic effects of cannabinoids and their role involving peripheral, spinal and supraspinal structures are well explained.

This review is very interesting and is easy to understand and makes all the concepts very clear.

There are minor revisions that I could suggest:

1)   in line 20 and 113 write the explanation of “CGRP” (calcitonin gene-related peptide)

2)   in line 39 write the explanation of TRPV1 as first time

3)   in line 45 correct “Δ9-tetrahydro cannabinol” in “Δ9-tetrahydrocannabinol”

4)   in line 48 correct “it” in “its”

5)   in line 59, 100, 174 correct “bene or to be” in “been”

6)   in my opinion the informations of TRP of line 134-140 could insert before in line 63 and also the sentences 140-151 could insert after line 67.

7)   in line 87, 138, 139, 141,334, 339 correct formatting

8)   in line 114 write the explanation of “SP”  (substance P) 

9)   in line 116 write the explanation of “DPN” (diabetic neuropathy)

10) in line 121 insert the the abbreviation (HIV-AN)

11) in line 134 delete “Transient Receptor Potential (TRP)” since it needs to write it before

12)  in line 155 delete “calcitonin gene-related peptide and substance P” since it needs to write it before

13) in line 160, 326 and 337 insert also the abbreviation in “nucleus tractus solitarius” (NTS)”

14) in line 167 delete the comma after “whereas” 

15) in line 171 delete “in femtomolar to picomolar ranges” since it is written before 

16) in line 200 correct “mechanism/s” 

17) in line 225 complete the concept (11% of ???migraines). 

18) in line 243, 246 and 250 write the explanation of “PPAR” (delete after in line 273), “PAG” (and delete after in line 252) and “GABA” respectively 

19)in line 254 delete “rostral ventromedial medulla” and use only abbrevation 

20)in line 268 use the abbrevation

21) in line 279 and 280, 375 use the abbrevation in “anandamide” and “fatty acid amide hydrolase” 

22) in line 286 insert the “serotonin” in “5-HT1A” 

23)  in line 318-320 “It is not recommended for use in pregnancy-induced nausea because the effects of cannabis on the unborn is not yet established”. In my opinion it is better to clarify this concept since THC interferes with the endocannabinoid system, which controls progenitor cell proliferation and neuronal differentiation, axon growth and synapse formation and pruning in the developing brain and a study demonstrated that prenatal cannabis exposure (PCE) predisposes to a wide array of behavioral and cognitive deficits including hyperactivity, enhanced impulsivity, loss of sustained attention, increased sensitivity to drugs of abuse and susceptibility to psychosis (see Frau R, Miczán V, Traccis F, Aroni S, Pongor CI, Saba P, Serra V, Sagheddu C, Fanni S, Congiu M, Devoto P, Cheer JF, Katona I, Melis M. Prenatal THC exposure produces a hyperdopaminergic phenotype rescued by pregnenolone. Nat Neurosci. 2019 Dec;22(12):1975-1985).

However, make the difference between THC and CBD, if it is present

24)in line 352, 383 insert “AP” and the explanation “neurokinin-1 or tachykinin” in “NK1” as reported on line 382

25) in line 387 insert the explanation “8-OH-DPAT” 

26)  correct formatting references

Author Response

Reviewer 2

We thank the reviewer for their useful comments on this manuscript. We have addressed all the issues to the best of our abilities point by point. I hope the reviewers find the answers adequate and the manuscript suitable for publication. We have included two files (changes shown and changes accepted) 

This review is very interesting and is easy to understand and makes all the concepts very clear.There are minor revisions that I could suggest:

  • in line 20 and 113 write the explanation of “CGRP” (calcitonin gene-related peptide)

The abbreviation CGRP has been expanded

  • in line 39 write the explanation of TRPV1 as first time

We have expanded the abbreviation TRPV1

  • in line 45 correct “Δ9-tetrahydro cannabinol” in “Δ9-tetrahydrocannabinol”

We have corrected.

  • in line 48 correct “it” in “its”

We have corrected.

  • in line 59, 100, 174 correct “bene or to be” in “been”

‘bene and been’ corrected to ‘been’

  • in my opinion the informations of TRP of line 134-140 could insert before in line 63 and also the sentences 140-151 could insert after line 67.

We have moved this section as suggested by the reviewer

  • in line 87, 138, 139, 141,334, 339 correct formatting

We have corrected the formatting.

  • in line 114 write the explanation of “SP”  (substance P) 

We have expanded the abbreviation.

  • in line 116 write the explanation of “DPN” (diabetic neuropathy)

DPN has been expanded as Diabetic Peripheral Neuropathy

  • in line 121 insert the the abbreviation (HIV-AN)

We have made the change.

  • in line 134 delete “Transient Receptor Potential (TRP)” since it needs to write it before

We have corrected.

  • in line 155 delete “calcitonin gene-related peptide and substance P” since it needs to write it before

We have corrected.

  • in line 160, 326 and 337 insert also the abbreviation in “nucleus tractus solitarius” (NTS)”

We have modified.

  • in line 167 delete the comma after “whereas” 

We have deleted.

  • in line 171 delete “in femtomolar to picomolar ranges” since it is written before 

We have corrected.

  • in line 200 correct “mechanism/s” 

We have corrected.

  • in line 225 complete the concept (11% of ???migraines). 

We have correct as “11% of migraineurs………..”

  • in line 243, 246 and 250 write the explanation of “PPAR” (delete after in line 273), “PAG” (and delete after in line 252) and “GABA” respectively.

We have corrected and expanded ‘GABA’ as gamma aminobutyric acid

19)in line 254 delete “rostral ventromedial medulla” and use only abbreviation.

      We have corrected this as ‘RVM’

20)in line 268 use the abbreviation

We have modified.

21) in line 279 and 280, 375 use the abbrevation in “anandamide” and “fatty acid amide hydrolase” 

      We have modified.

22) in line 286 insert the “serotonin” in “5-HT1A” 

      We have modified.

23)  in line 318-320 “It is not recommended for use in pregnancy-induced nausea because the effects of cannabis on the unborn is not yet established”. In my opinion it is better to clarify this concept since THC interferes with the endocannabinoid system, which controls progenitor cell proliferation and neuronal differentiation, axon growth and synapse formation and pruning in the developing brain and a study demonstrated that prenatal cannabis exposure (PCE) predisposes to a wide array of behavioral and cognitive deficits including hyperactivity, enhanced impulsivity, loss of sustained attention, increased sensitivity to drugs of abuse and susceptibility to psychosis (see Frau R, Miczán V, Traccis F, Aroni S, Pongor CI, Saba P, Serra V, Sagheddu C, Fanni S, Congiu M, Devoto P, Cheer JF, Katona I, Melis M. Prenatal THC exposure produces a hyperdopaminergic phenotype rescued by pregnenolone. Nat Neurosci. 2019 Dec;22(12):1975-1985).

However, make the difference between THC and CBD, if it is present

We really appreciate pointing out this issue, we have modified the text to reflect the reviewers concern. We have included this reference.

24)in line 352, 383 insert “AP” and the explanation “neurokinin-1 or tachykinin” in “NK1” as reported on line 382

      We have modified.

25) in line 387 insert the explanation “8-OH-DPAT” 

      We have expanded the abbreviation.

26)  correct formatting references

      We have formatted the references.
